# In vitro FRET analysis of IRE1 and BiP association and dissociation upon endoplasmic reticulum stress

Megan C Kopp, Piotr R Nowak, Natacha Larburu, Christopher J Adams, Maruf MU Ali*

Department of Life Sciences, Centre for Structural Biology, Imperial College London, London, United Kingdom

**Abstract** The unfolded protein response (UPR) is a key signaling system that regulates protein homeostasis within the endoplasmic reticulum (ER). The primary step in UPR activation is the detection of misfolded proteins, the mechanism of which is unclear. We have previously suggested an allosteric mechanism for UPR induction (*Carrara et al., 2015*) based on qualitative pull-down assays. Here, we develop an in vitro Förster resonance energy transfer (FRET) UPR induction assay that quantifies IRE1 luminal domain and BiP association and dissociation upon addition of misfolded proteins. Using this technique, we reassess our previous observations and extend mechanistic insight to cover other general ER misfolded protein substrates and their folded native state. Moreover, we evaluate the key BiP substrate-binding domain mutant V461F. The new experimental approach significantly enhances the evidence suggesting an allosteric model for UPR induction upon ER stress.
DOI: https://doi.org/10.7554/eLife.30257.001

## Introduction

The unfolded protein response is a key signaling system that regulates protein homeostasis within the ER. The response is induced when there is an accumulation of misfolded protein—due to increase in protein load or aberrant protein folding, which results in activation of a cellular program that aims to restore correctly folded protein levels, in order to ensure a properly functioning ER (*Schneider and Bertolotti, 2015*; *Wang and Kaufman, 2016*; *Hetz and Papa, 2017*).

A critical step in this process is the initial detection of misfolded proteins that leads to UPR induction, the molecular mechanism of which is unclear. We have previously reported an allosteric model for UPR induction (*Carrara et al., 2015*). Our model was based on the observation that a interaction between the luminal domain (LD) of the key UPR protein, IRE1, and the ATPase domain of BiP, an ER Hsp70 chaperone, dissociates upon the binding of $C_H1$ misfolded protein to the canonical BiP substrate-binding domain. To observe this important step, we utilized a pull-down assay that qualitatively measured noncanonical dissociation (*Carrara et al., 2015*). In the present study, we developed a Förster resonance energy transfer (FRET) UPR induction assay that quantifies the association and subsequent dissociation of IRE1 LD with BiP, upon addition of misfolded protein. This new experimental technique reconstitutes in vitro the most crucial mechanistic step in UPR signaling, namely, the detection of misfolded protein and UPR induction. We reassess our previous observations, and extend our measurements to other general ER misfolded proteins and their native folded states. Moreover, we evaluate the important BiP substrate-binding domain mutant $BiP^{V461F}$, in the presence of nucleotide and in combination with a mutation that renders BiP ATPase deficient. The new experimental approach significantly enhances the evidence to suggest an allosteric model for UPR induction upon ER stress.

**\*For correspondence:**
maruf.ali@imperial.ac.uk

**Competing interests:** The authors declare that no competing interests exist.

## Results

In order to generate a quantitative measure of association and dissociation between IRE1 LD and BiP, we decided to make use of FRET between two fluorophores: cyan fluorescent protein (CFP) and yellow fluorescent protein (YFP) (*Martin et al., 2008*; *Pollok and Heim, 1999*; *Felber et al., 2004*; *Bajar et al., 2016*).

As we have previously measured an interaction between the N-terminal ATPase domain of BiP and IRE1 LD ($K_d$ = 1.33 μM), and our pull-down assays utilized an N-terminus-positioned affinity tag that had no impact on interaction (*Carrara et al., 2015*), we attached fluorescent proteins to the N-terminus of both constructs, with YFP connected via short linker to BiP, upstream of the ATPase domain; and similarly, CFP to IRE1 LD (*Figure 1A*).

Measurement of an interaction using this technique would require excitation of CFP at 430 nm wavelength, and observation of emission at the longer wavelength of 480 nm. If YFP is in close proximity, approximately 1–10 nm distance (*Bajar et al., 2016*), then energy transfer will occur between CFP and YFP, resulting in an emission peak at 530 nm wavelength. This scenario will only occur if there is a direct interaction between the IRE1 LD and the BiP ATPase domain that brings the fluorescent proteins into close proximity (*Figure 1B*). For our assay, we excited at 430 nm wavelength with bandwidth 10 nm (430-10 nm), and observed the fluorescence emission intensity at 530 nm wavelength with a bandwidth of 10 nm (530-10 nm), which was then divided by the emission intensity at 480 nm also with bandwidth 10 nm (480-10 nm) to give a FRET ratio (530-10 nm/480-10nm), and this was used to measure signal output. This ratio-metric measurement produces less noise then individual fluorescent intensity observations, and is a more reliable FRET signal measure (*Bajar et al., 2016*; *Pollok and Heim, 1999*; *Martin et al., 2008*). We mixed CFP-IRE1 LD and YFP-BiP protein in equimolar ratio, and compared it to non-binding controls. We used two non-binding controls: YFP with CFP-IRE1 LD, and BiP-YFP (with YFP positioned at the C-terminal, beyond the substrate-binding domain) with CFP-IRE1 LD.

Upon excitation, we observed a FRET ratio (530-10 nm/480-10nm) of ~0.62 for CFP-IRE1 LD with YFP-BiP. For our non-binding controls, we measured a FRET ratio of ~0.34 (*Figure 1C*). The data show an almost doubling of the FRET ratio for CFP-IRE1 LD with YFP-BiP when compared to non-binding controls. This represents a significant FRET signal between the two fluorescent proteins (*Pollok and Heim, 1999*) that was easily reproducible. Moreover, it indicates a direct interaction between the IRE1 LD and the BiP ATPase domain that we can quantifiably measure.

We have previously demonstrated that misfolded protein, $C_H1$, binds to the canonical SBP of BiP, and effects dissociation of BiP from IRE1 LD. We reasoned that the addition of $C_H1$ to our assay should cause significant reduction in the FRET signal. To this end, we added 10-fold molar excess of $C_H1$ to CFP-IRE1 LD and YFP-BiP, and measured the FRET signal ratio. To make interpretation easier, we took the CFP-IRE1 LD with YFP-BiP sample to represent 100% FRET signal, and the non-binding controls to represent 0% FRET signal. Upon addition of $C_H1$, the FRET signal was reduced to ~4% (*Figure 1D*), clearly indicating that the binding of $C_H1$ misfolded protein to the canonical BiP SBD caused the dissociation of the BiP ATPase domain from the IRE1 LD.

In order to give us a better understanding of $C_H1$'s ability to inhibit the FRET signal, we added varying concentrations of $C_H1$ to CFP-IRE1 LD and YFP-BiP, and measured the FRET signal. We found that the addition of $C_H1$ reduced the FRET signal in a dose-dependent fashion (inhibition constant $K_i$ = 0.51 ± 0.01 μM) (*Figure 2*), with 10-fold molar excess reducing the FRET signal to almost non-binding control levels. This suggests that a 10-fold molar excess of $C_H1$ was sufficient for almost complete inhibition of FRET signal.

To confirm that the FRET signal was a consequence of IRE1 binding to the BiP ATPase domain, we generated YFP-BiP ATPase domain and YFP-BiP SBD proteins (*Figure 3A*), and measured the FRET signal upon their interaction with CFP-IRE1 LD. We observed a 100% FRET signal equivalent to that resulting from full-length YFP-BiP interaction with CFP-IRE1 LD, whereas we measured no FRET signal when using the SBD (*Figure 3B–C*). This reaffirms the notion that the ATPase domain of BiP is necessary for the interaction with IRE1 LD, consistent with microscale thermophoresis (MST) and pull-down assay measurements (*Carrara et al., 2015*); furthermore, this interaction has also been suggested by another study (*Todd-Corlett et al., 2007*).

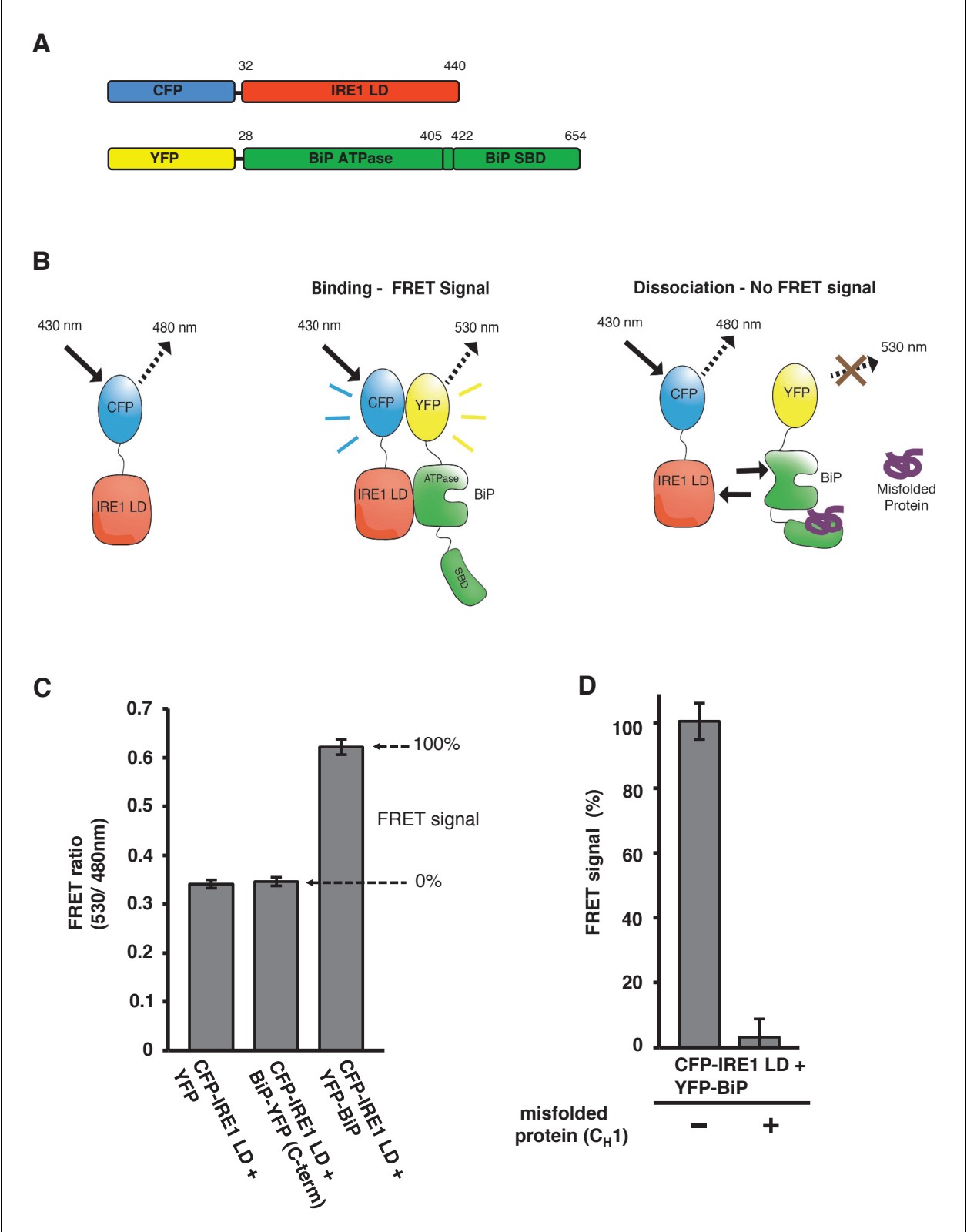

**Figure 1.** In vitro FRET assay measures noncanonical IRE1 and BiP association and dissociation. (A) Schematic description of constructs used for the assay, with numbers denoting the amino acid residue of the protein. Both fluorescent proteins, CFP and YFP, are attached N-terminally to the IRE1 LD and BiP proteins, respectively. (B) Schematic representation of the in vitro FRET UPR induction assay, which measures a three-component protein interaction system. CFP-IRE1 LD will be excited at 430 nm with a bandwidth of 10 nm (430–10 nm), and upon excitement it will emit radiation at the

*Figure 1 continued on next page*

*Figure 1 continued*

longer wavelength of 480–10 nm. When YFP-BiP is added, IRE1 LD should interact with BiP via its ATPase domain, bringing YFP in close proximity to CFP, resulting in FRET between the fluorescent proteins and an emission at 530–10 nm. Dissociation of the complex—with loss of FRET signal—should occur upon addition of misfolded protein, which will bind to BiP substrate-binding domain (SBD) to cause conformational change. The ratio of 530-10 nm/480-10 nm will be used to measure FRET signal output. (C) Bar graph of the FRET ratio (530–10 nm/480–10 nm) upon excitation at 430–10 nm wavelength when CFP-IRE1 LD and YFP-BiP were mixed in equimolar amounts. This was compared to non-binding controls, YFP with CFP-IRE1 LD, and BiP with C-terminally tagged YFP with CFP-IRE1 LD. The FRET ratio was almost doubled upon interaction revealing a clear FRET signal. The negative controls measure a FRET ratio of ~0.34 due to CFP, which contributes a significant fluorescence emission intensity at 530 nm (also referred to as CFP leakage) when excited at 430 nm. This allows for greater spectral overlap with YFP making CFP and YFP excellent FRET pairs, but adds to the background noise. The data are shown as mean ± SD (n = 6). (D) FRET UPR induction assay measurements upon addition of misfolded protein $C_H1$. In this graph, the FRET signal is represented as a percentage, with 0% observation equivalent to non-binding control and 100% being represented by CFP-IRE1 LD and YFP-BiP. The addition of $C_H1$ caused IRE1 LD and BiP dissociation, resulting in the loss of FRET signal (mean ± SD; n = 6).
DOI: https://doi.org/10.7554/eLife.30257.002

The following source data is available for figure 1:

**Source data 1.**
DOI: https://doi.org/10.7554/eLife.30257.003

In addition, we previously demonstrated that non-canonical IRE1 LD and BiP interaction was independent of nucleotides, and hence BiP chaperoning activity. We tested this observation again using our FRET UPR induction assay. We added 5 mM ATP, ADP and AMP-PNP to CFP-IRE1 LD and YFP-BiP, and compared these to samples examined in the absence of nucleotide. We observed no significant difference upon the addition of various nucleotides, or in the absence of nucleotide, with all

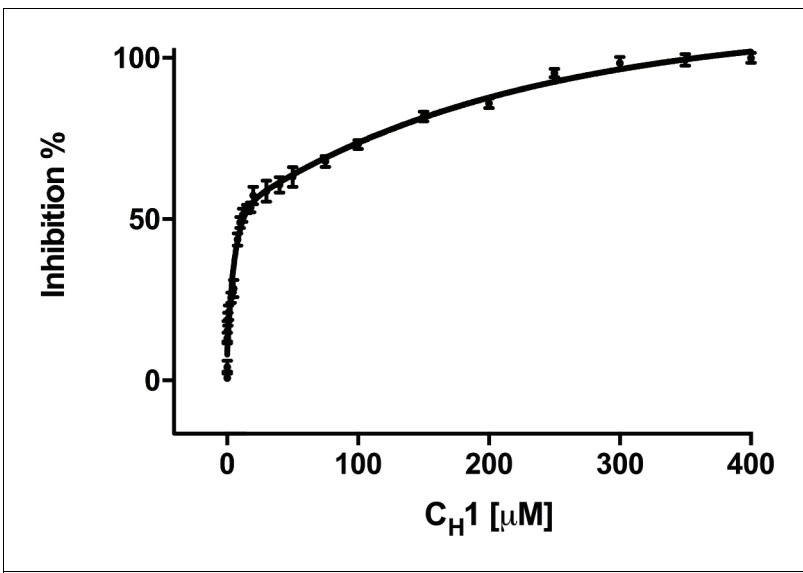

**Figure 2.** Addition of misfolded protein $C_H1$ inhibits FRET signal in a dose-dependent fashion. Graph showing the percentage inhibition of FRET signal as a function of the concentration of $C_H1$ (mean ± SD; n = 3). The binding of $C_H1$ to the YFP-BiP caused its dissociation from the CFP-IRE1 LD. At 300 μM $C_H1$, a 10-fold molar excess over both CFP-IRE1 LD and YFP-BiP, the signal was almost completely inhibited. Fitting an exponential (two phase association, $r^2$ = 0.98) curve to data gave an $IC_{50}$ of 12.3 ± 1.3 μM, the amount of $C_H1$ required to give 50% dissociation, with an inhibition constant $K_i$ = 0.51 ± 0.01 μM. The inhibition constant $K_i$ is the binding constant that relates the interaction between a protein complex with a binding affinity $K_d$ and an inhibiting molecule that is derived from $IC_{50}$ (**Martin et al., 2008**).
DOI: https://doi.org/10.7554/eLife.30257.004

The following source data is available for figure 2:

**Source data 1.**
DOI: https://doi.org/10.7554/eLife.30257.005

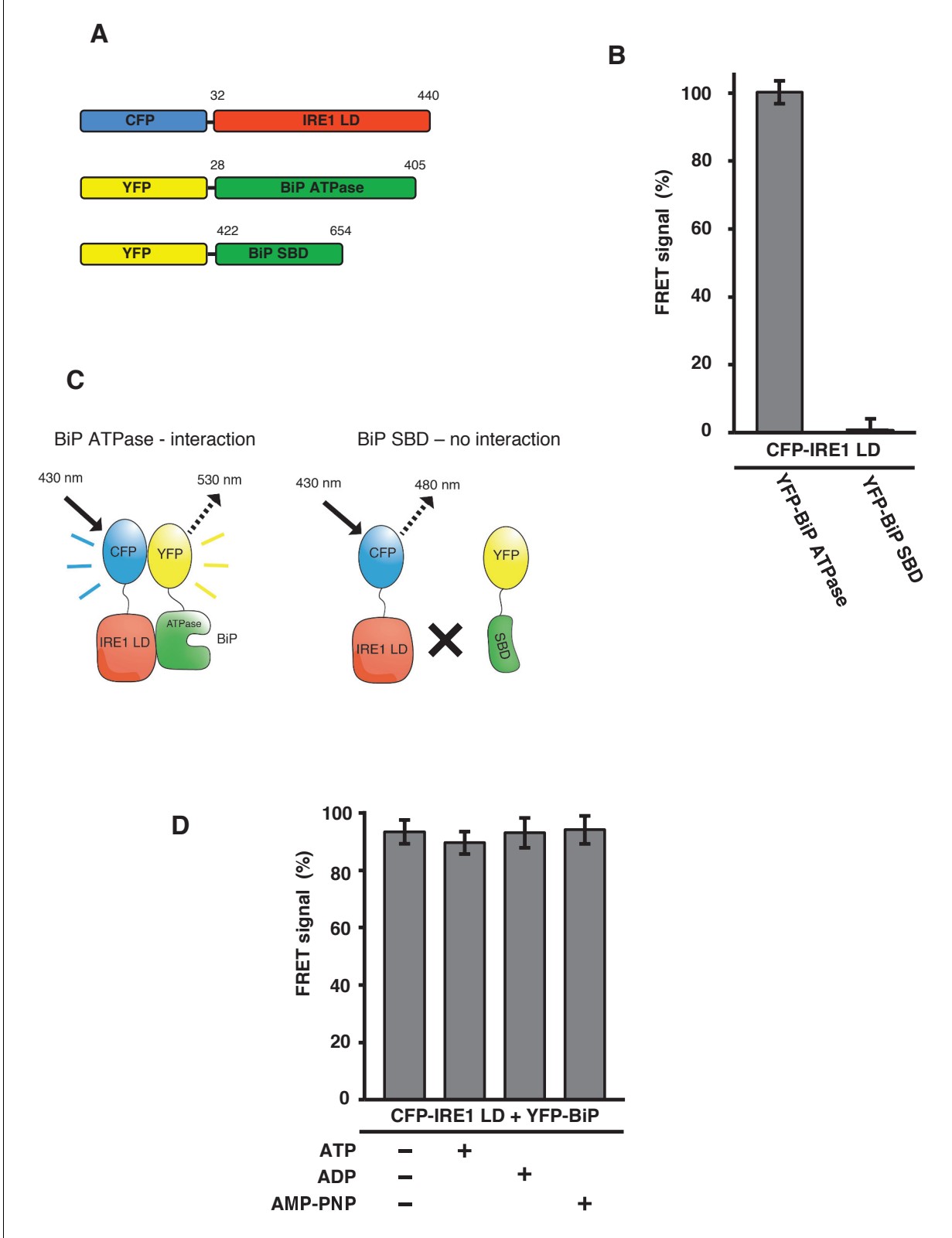

**Figure 3.** IRE1 LD interacts with the BiP ATPase domain to produce FRET signal, independent of nucleotides. (**A**) Diagram detailing the BiP ATPase and SBD constructs used for measuring the interaction with IRE1 LD. (**B**) IRE1 LD interacted with the BiP ATPase domain to produce 100% FRET signal equivalent to that produced by full-length BiP. No FRET signal observed with the SBD (mean ± SD; n = 6). (**C**) A schematic illustrating that there was no observable interaction between SBD and IRE1 LD, with no corresponding FRET signal. (**D**) 5 mM ATP, ADP and AMP-PNP were added to CFP-IRE1 LD

*Figure 3 continued on next page*

*Figure 3 continued*

and YFP-BiP samples and the FRET signal was analyzed and compared to 100% FRET signal. The addition of nucleotides did not have a significant impact upon the FRET signal and the interaction between IRE1 and BiP.

DOI: https://doi.org/10.7554/eLife.30257.006

samples measuring almost 100% FRET signal (*Figure 3D*). This result again suggests that IRE1 LD–BiP association is independent of nucleotide binding.

To gain further mechanistic insights, we wanted to address whether other ER misfolded proteins could cause dissociation of BiP from IRE1; thereby, applying our model to other general ER misfolded proteins. A literature survey identified a number of ER misfolded protein substrates that had been used previously, including α1-antitrypsin (*Tsutsui and Wintrode, 2007*), RNase A (*Petrova et al., 2008*), apolipoprotein (*Morrow et al., 1999*), and transthyretin mutant D18G (TTR$^{D18G}$) (*Sato et al., 2007*). Interestingly, some of these misfolded protein substrates were generated from their correctly folded state in vitro. This allowed us to evaluate not only the misfolded protein, but also the same proteins in their folded native state, thus providing an important control that we were previously unable to assess using the inherently misfolded protein C$_H$1. We added both folded and misfolded proteins to our assay, and compared output to 100% FRET signal sample and 0% non-binding control. The addition of folded antitrypsin, RNase A, and apolipoprotein had no affect on the FRET signal, with all three measurements close to or equal to 100% (*Figure 4A–C*). The addition of misfolded versions of the same ER proteins: antitrypsin, RNase A, and apolipoprotein, reduced the signal to ~6% (*Figure 4A–C*). For TTR$^{D18G}$, which lacked a folded control, the FRET signal was reduced to ~2% (*Figure 4—figure supplement 1*). First, the data clearly indicate that only the misfolded state of the ER proteins induce an effect, by binding to canonical BiP SBD, resulting in IRE1 and BiP dissociation via the ATPase domain, with resultant loss in FRET signal. Second, dissociation of IRE1 and BiP upon binding of misfolded protein is not specific to C$_H$1 substrate, rather other ER misfolded proteins can induce dissociation by binding to BiP SBD, consistent with BiP being a HSP70-type chaperone.

To reinforce these observations, we repeated the experiment utilizing our previously reported qualitative pull-down assay, in which GST- or His$_6$-tagged BiP forms a complex with IRE1 LD that was visualized on aSDS-PAGE gel. In the presence of native folded ER proteins—antitrypsin, RNase A, or apolipoprotein—the interaction between IRE1 LD and BiP remained intact. Upon addition of misfolded versions of the same ER proteins, the interaction between the IRE1 LD and the ATPase domain of BiP dissociated, leaving BiP bound to the misfolded protein via its SBD (*Figure 4D–F*). Therefore, the qualitative pull-down assay reproduced the observations made with the FRET UPR induction assay.

V461F is a key mutation of BiP that prevents misfolded substrate engagement (*Laufen et al., 1999*; *Petrova et al., 2008*); it does this by inserting a bulky side chain that sterically hinders access to the active site of the SBD. Our allosteric model predicts that dissociation of the IRE1 LD and the ATPase of this mutated BiP will not occur upon addition of misfolded protein, with no subsequent loss in the FRET signal. We generated YFP-BiP$^{V461F}$ protein, and compared it to YFP-BiP$^{WT}$ by observing the FRET signal in the absence and presence of misfolded protein. For YFP-BiP$^{WT}$, we observed a 95% reduction in the FRET signal upon addition of misfolded protein. By contrast, the addition of misfolded protein to the YFP-BiP$^{V461F}$ sample showed no significant loss in FRET signal (*Figure 5A–C*). Furthermore, to emphasize the point that the interaction between the IRE1 LD and BiP was not regulated by ATP, but that the primary determinant for dissociation was misfolded proteins, we observed the FRET signal upon the addition of ATP and using BiP$^{V461F}$ in combination with a mutation that renders BiP deficient in its ATPase activity, T229A (*Gaut and Hendershot, 1993*; *Petrova et al., 2008*). In the presence of 5 mM ATP, both BiP$^{WT}$ and BiP$^{T229A}$ interacted with CFP-IRE1 LD; moreover, dissociation of complex was caused by addition of misfolded protein in a manner that was independent of ATP (*Figure 5—figure supplement 1*). However, the double mutant, BiP$^{V461F, T229A}$, associated with CFP-IRE1 LD in the presence of ATP, but failed to dissociate upon the addition of misfolded protein (*Figure 5—figure supplement 1*). Thus this mutant behaved in exactly the same way as the SBD single mutant, BiP$^{V461F}$ (*Figure 5A–C*), indicating again that the BiP–IRE1 association and dissociation were independent of nucleotides..

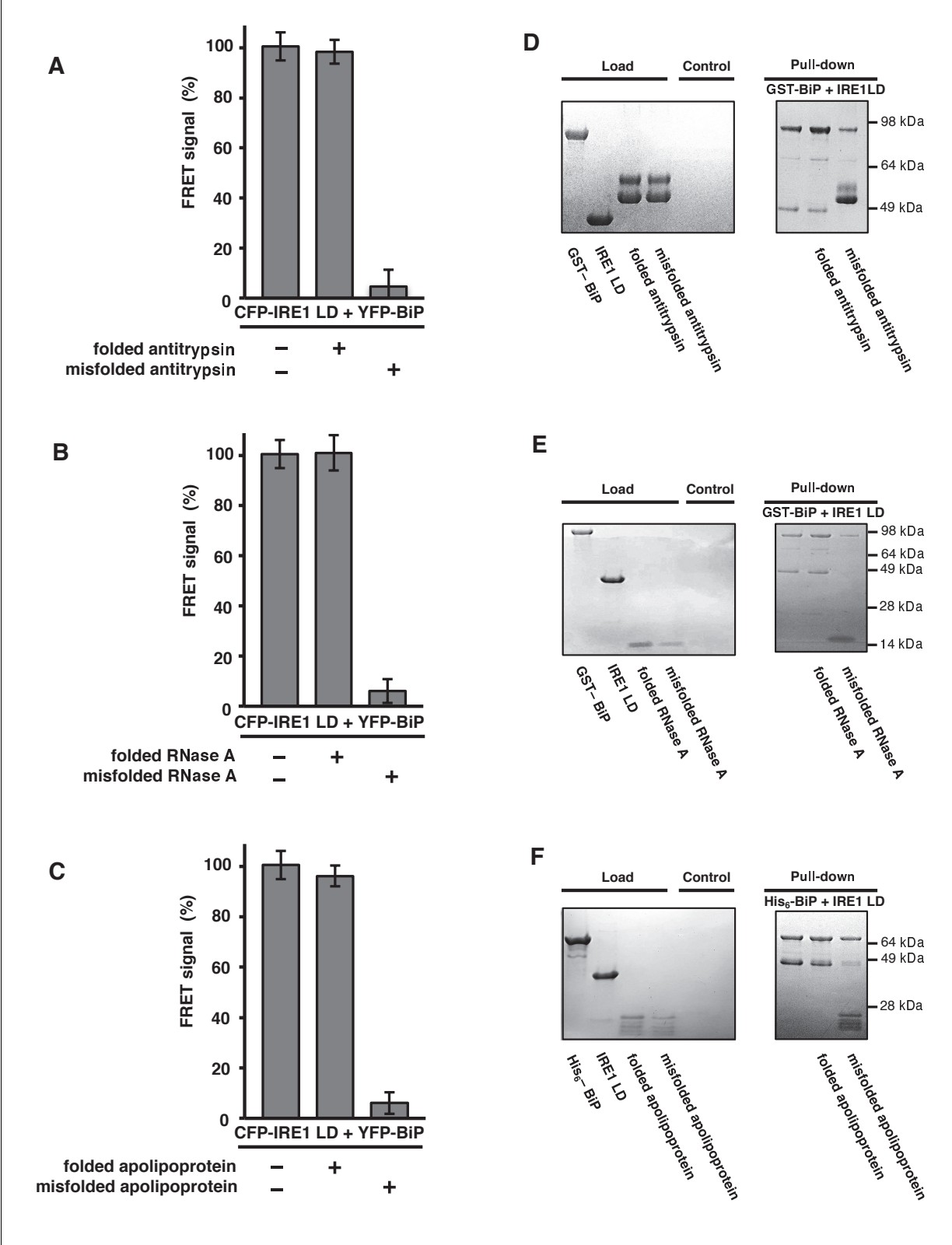

**Figure 4.** General ER misfolded proteins, – but not their folded state, cause dissociation. (A–C) Measurements from a FRET UPR induction assay upon addition of folded and misfolded ER proteins. (A) Antitrypsin. (B) RNase A. (C) Apolipoprotein. All data shown are mean ± SD (n = 6). (D–F) Qualitative pull-down assays, stained with coomassie brilliant blue, showed IRE1 LD and BiP dissociation upon binding of a misfolded version of an ER protein. Dissociation did not occur upon the addition of folded proteins. (D) Antitrypsin. (E) RNase A. (F) Apolipoprotein.

*Figure 4 continued on next page*

*Figure 4 continued*

DOI: https://doi.org/10.7554/eLife.30257.007

The following figure supplement is available for figure 4:

**Figure supplement 1.** TTR$^{D18G}$ affects IRE1 LD and BiP dissociation.

DOI: https://doi.org/10.7554/eLife.30257.008

Next, to support our in vitro data, we attempted to measure an interaction in cells using co-immunoprecipitation. We co-transfected HEK293 cells with IRE1 and either BiP wildtype or BiP V461F in the absence and presence of ER stress. In the absence of ER stress, we observed an interaction both between IRE1 and BiP wildtype and between IRE1 and BiP V461F. Upon addition of ER stress, however, the interaction between IRE1 and BiP wildtype was significantly reduced, whereas no loss in interaction was observed with BiP V461F, thus supporting the in vitro FRET data. Taken together, both the in vitro and cellular data indicate that when BiP forms an interaction with the IRE1 LD, via its ATPase domain, misfolded proteins can bind to the canonical active site of the BiP SBD to cause dissociation of BiP, resulting in loss of FRET signal. The BiP$^{V461F}$ mutant, in which access to the active site is blocked, was unable to interact with misfolded protein and hence could not effect the dissociation of BiP from IRE1, resulting in no reduction of FRET signal. Moreover, as misfolded protein binding to the BiP substrate-binding site caused BiP ATPase domain to dissociate from IRE1, an allosteric mechanism is clearly suggested, supporting our model.

## Discussion

In the present study, we have developed an in vitro FRET UPR induction assay that quantifies the association and dissociation of the IRE1 LD from BiP upon ER stress; thereby, describing an important technical tool for dissecting the UPR induction mechanism that measures a three-component interacting system. We extend our observations to cover other general ER misfolded proteins and make comparisons to these proteins in their native folded state. We also evaluate the key BiP SBD mutant BiP$^{V461F}$, in the presence of ATP and in combination with an ATPase-deficient BiP mutant. The mutant data, along with the observation that ER misfolded proteins (and not the same proteins in their native folded state) cause dissociation, clearly indicates that misfolded proteins are being recognized or sensed by the BiP SBD when interacting with the IRE1 LD. Interestingly, the present data provide evidence that would discount a BiP competition model (*Bertolotti et al., 2000*; *Kimata et al., 2003*; *Okamura et al., 2000*; *Liu et al., 2000*). In this model, IRE1 binds to BiP via the SBD, which is the same site that misfolded proteins bind to BiP. Upon ER stress, misfolded protein titre BiP off from IRE1 in a competitive fashion, resulting in dimerization and activation of UPR. This process is ATP dependent (*Bertolotti et al., 2000*; *Kimata et al., 2003*), with ATP (and not necessarily misfolded proteins) being sufficient to cause dissociation of IRE1 and BiP. However, as the interaction between IRE1 and BiP is mediated via the SBD, this suggests a chaperone-substrate type interaction (*Bertolotti et al., 2000*; *Kimata et al., 2003*); this point is further emphasized by the process being ATP dependent—ATPase activity is an integral part of the Hsp 70 chaperone-substrate mechanism.

Our in vitro data support an allosteric model (*Carrara et al., 2015*); whereby, the interaction between IRE1 and BiP occurs via the ATPase domain of BiP and is independent of nucleotides; this clearly indicates a UPR signaling productive association and not a chaperone-substrate type interaction. Furthermore, the primary determining factor for dissociation is misfolded protein binding to the SBD of BiP, which engenders an allosteric change that results in the dissociation of the ATPase domain of BiP from IRE1 (*Carrara et al., 2015*). Moreover, as binding between IRE1 and BiP, and the detection of misfolded proteins, involves two different domains of BiP that are coupled by conformational change, there is no requirement for the association with IRE1 and detection of misfolded protein to be a competitive process, thus reconciling the sensitive nature of UPR signaling.

Although our model indicates that the binding and release of BiP from IRE1 is independent of ATPase activity, it could be that nucleotides have influence over other aspects of UPR signaling. It is known that nucleotides greatly impact BiP conformation, with ADP-bound BiP adopting a

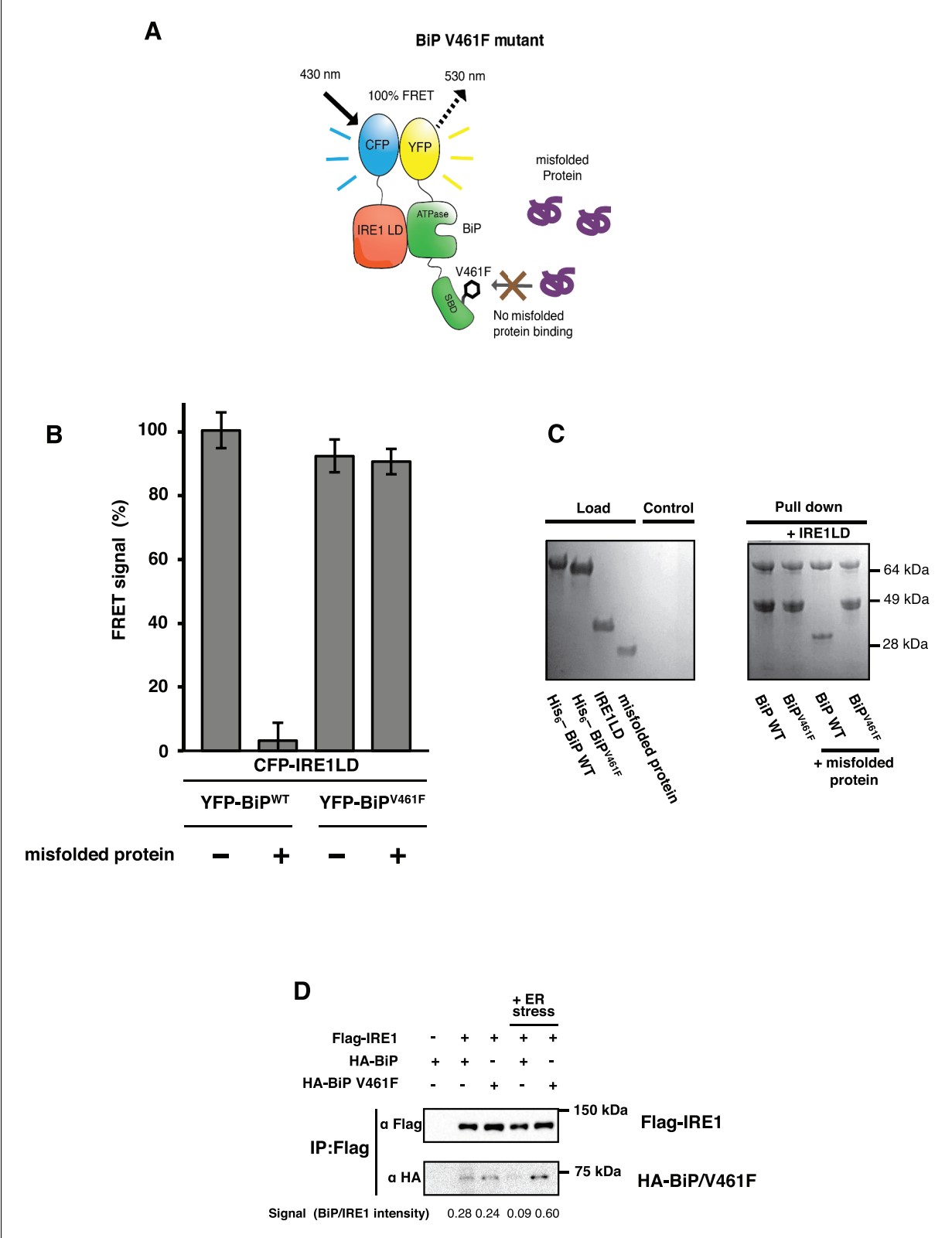

**Figure 5.** BiP V461F mutation prevents dissociation of the IRE1 LD and BiP upon addition of misfolded protein. (**A**) A schematic illustrating the location of the V461F mutation within the SBD of BiP, which is represented by a six-sided black ring that denotes the phenylalanine residue. If the mutation prevents the binding of misfolded protein to BiP, then there will be no allosteric change, and no dissociation of the IRE1–BiP complex, with the FRET signal remaining intact. (**B**) The FRET signal was significantly reduced upon addition of misfolded protein to BiP^WT–IRE1 LD sample. By contrast, there

*Figure 5 continued on next page*

*Figure 5 continued*

was no loss of FRET signal with BiP V461F. (**C**) A secondary pull-down assay recapitulates the FRET assay results, with BiP[WT] fully able to respond to misfolded protein, whereas BiP V461F was unaffected by the presence of misfolded protein. (**D**) Various combinations of Flag-IRE1, HA-BiP and HA-BiP V461F were co-expressed in HEK293 cells and treated with 5 mM tunicamycin to replicate ER stress. Flag-IRE1 was immunoprecipitated with anti-Flag magnetic resin and then samples were immunoblotted with both Flag and HA antibodies, before and after the addition of ER stress.

DOI: https://doi.org/10.7554/eLife.30257.009

The following figure supplement is available for figure 5:

**Figure supplement 1.** Misfolded proteins are the primary determinant for dissociation of the IRE1 LD and BiP.

DOI: https://doi.org/10.7554/eLife.30257.010

substantially different conformation to that of ATP-bound BiP. Exactly how this relates to UPR signaling is an area of ongoing research.

The results of the current study are also consistent with our previous data from microscale thermophoresis (MST) and pull-down assays that have shown an interaction between the ATPase domain (and not the SBD) of BiP and IRE1; this observation has been reported before (*Todd-Corlett et al., 2007*). In summary, we develop an in vitro FRET UPR induction assay that quantifies non-canonical association and dissociation upon ER stress. Furthermore, the data enhance the experimental evidence suggesting an allosteric mechanism for UPR induction.

## Materials and methods

### Expression and purification of IRE1 LD and BiP fluorescent fusion proteins

Fluorescent fusion constructs were designed in which the fluorescent proteins CFP and YFP were attached N-terminally to human IRE1 LD (32-440) and human BiP (28-654), respectively, via a short 15aa linker (GGAGGAGGAGGAGGA). This construct was then cloned into a pGEX 6p1 vector, containing a GST tag and a PreScission protease site, and expressed in *Escherichia coli* BL21 (DE3) cells (Agilent, CA, USA). Separately, non-fluorescent human BiP and IRE1 LD proteins were cloned and expressed with either an N-terminal His$_6$-tag or a GST-tag followed by a PreScission Protease cleavage site. His$_6$-tagged proteins were purified by Co$^{2+}$-NTA affinity using TALON metal affinity resin (Clontech, CA, USA) in buffer A (50 mM HEPES [pH 8.0], 400 mM NaCl and 10% glycerol) and eluted in the presence of 250 mM imidazole. GST-tagged proteins were purified using Agarose Glutathione Affinity resin (ThermoFisher, MA, USA) in buffer B (50 mM HEPES [pH 7.3], 400 mM NaCl, 1 mM DTT, and 10% glycerol) and eluted in buffer A supplemented with 10 mM reduced glutathione and 1 mM DTT. Initial lysis and purification steps for BiP were supplemented with 5 mM ATP and 10 mM MgCl$_2$. Unless otherwise specified, the His$_6$-tag or GST-tag was removed by overnight incubation with PreScission Protease followed by an additional affinity step to remove uncleaved protein. All proteins were further purified by anion-exchange using a HiTrap Q HP column (GE Healthcare, UK) and size-exclusion chromatography on a HiLoad 16/60 Superdex 200 column in buffer C (50 mM HEPES [pH 8.0], 50 mM NaCl, 10 mM KCl, 10% glycerol).

### Expression and purification of misfolded proteins

Soluble RNaseA, $\alpha_1$-Anti-trypsin, and Apolipoprotein-AI were purchased from Sigma (MO) and then unfolded for use in experiments. Folded versions of $\alpha_1$-Anti-trypsin, Apolipoprotein-AI, and RNaseA were dissolved in buffer C with no other treatment. For misfolded proteins, $\alpha_1$-Anti-trypsin and Apolipoprotein-AI were dissolved in buffer C and heated at 60°C for 1 hr. RNaseA was incubated in 6 M Guanidinium-HCL and 0.1 M DTT overnight then dialyzed into buffer C. C$_H$1 protein was expressed as previously described (*Marcinowski et al., 2011*). TTR mutant D18G was expressed as a fusion protein with an N-terminal GST-tag, which was not removed, and purified as described above for GST-tagged proteins.

### FRET assay

FRET assay experiments were carried out using a CLARIOstar microplate reader (BMG Labtech, DE). Fluorescently labelled proteins were combined in an equimolar ratio in all samples. Folded or

misfolded protein was added at 10-fold molar excess to the concentration of the fluorescent proteins, unless the concentration is otherwise specified. The mixture was then diluted with buffer C to a volume of 160 µL, adjusting the concentration for each of the fluorescent proteins to 30 µM. Where stated, buffer C was supplemented with 5 mM ATP, ADP, or AMP-PNP and 10 mM $MgCl_2$. 50 µL of sample was loaded into each well of a 384-well microplate (Greiner Bio-one, AT). The sample was excited at 430–10 nm wavelength, and the fluorescence emission intensity at 530–10 nm divided by the emission intensity at 480–10 nm (to give a FRET ratio) was used to measure FRET signal. Experiments were conducted as n = 6 and presented as mean ± SD, unless otherwise stated. To deduce the inhibition constant $K_i$ by measuring the loss of FRET signal by a tertiary protein, the protocol described by *Martin et al. (2008)* was followed using a $C_H1$ concentration ranging from 0.001 µM to 400 µM to calculate $IC_{50}$ with nonlinear regression performed in graphpad6. The other variables used for deducing $K_i$ were: initial substrate concentration of 30 µM and $K_d$ = 1.33 µM.

## Pull-down assay

All pull-down experiments were conducted in 2 mL gravity flow columns. 75 µL of TALON or Glutathione resin pre-equilibrated with buffer C was incubated with 40 µM $BiP_{His}$ or $BiP_{GST}$, respectively. The resin was washed with 500 µL of buffer C to remove any unbound BiP. Then 50 µL of purified IRE1 LD at 150 µM was added and the mixture was incubated for 1 hr at room temperature (RT). The resin was then washed with 500 µL of Buffer C in 125 µL increments. Then 50 µL of 300 µM folded if applicable and unfolded RNase A, $\alpha_1$-Anti-trypsin, Apolipoprotein-AI, and TTR were added to the resin and incubated for 1 hr at RT. The resin was washed as previously described with buffer C, and subsequently re-suspended with 75 µL of buffer. Samples of the re-suspended resin were analysed on a 4–12% gradient SDS-PAGE gel.

## Cell culture and co-immunoprecipitation

Suspension Human Embryonic Kidney cells (Expi293F) were cultured in serum-free Gibco's Expi293 Expression medium. A day before transfection, 30,000,000 cells (15 ml) were split into 125 ml Erlenmeyer vented flasks and incubated in 37°C in 8% $CO_2$ atmosphere. The following day, cells were diluted to the concentration of $2.5 \times 10^6$ cells/ml and co-transfected. DNA containing Flag-IRE1 and either HA-BiP WT or HA-BiPV461F (which were present in pcDNA 3.1(+) vector) were co-transfected with ExpiFectamine reagent to a concentration of 15 µg total DNA (1 ug/ml DNA) (ThermoFisher), according to the manufacturer's protocol (ThermoFisher). After 48 hr, cells were either harvested or 5 µM tunicamycin was added for 2 hr to produce ER-stressed samples. Next, cells were lysed by FastPrep-24 5Ghomogenizer/Lysing Matrix D (MP Biomedicals) in 500 µl buffer containing 50 mM Tris pH 7.5, 150 mM NaCl, 1% digitonin and EDTA-free protease inhibitor cocktail (Sigma). Lysates were centrifuged (15 min, 20,000xg), diluted and mixed with equilibrated anti-Flag M2 magnetic beads (Sigma) and incubated at RT for 30 min. Next, magnetic beads were collected by magnet and washed three times in buffer containing 50 mM Tris pH 7.5, 150 mM NaCl, 0.2% digitonin and 1x protease inhibitor cocktail. Flag-tagged IRE1 was eluted from beads by competitive elution buffer composed of 50 mM Tris pH 7.5, 150 mM NaCl, 0.2% digitonin, 1x EDTA-free protease cocktail and 200 ug 3X Flag peptide (Sigma). Eluates were mixed with Laemmli buffer, boiled and run on Tris-Glycine Wedge 10% gel (ThermoFisher).

## Immunoblotting

Gels were transferred to nitrocellulose membrane (iBlot2 system, ThermoFisher) and blocked in TBST buffer plus 5% Marvel dried milk for 1 hr at RT. Next, anti-Flag, 1:1000 (Sigma) and anti-HA, 1:2000 (ThermoFisher) were added to blocking buffer (TBST + 1% milk powder) and incubated overnight at 4°C. Next, membranes were washed three times in TBST buffer and incubated with secondary antibody in TBST + 2% milk: anti-mouse-HRP, 1:6000 (GE Healthcare). After 1 hr incubation at RT and three further washes, blots were visualized by Millipore Luminata Crescendo Western HRP substrate and developed by the ChemiDoc MP gel imaging system (BioRad).

## Acknowledgements

We thank Himani Amin with help with protein purification. This work was funded by a Senior Cancer Research Fellowship awarded to MMUA (C33269/A20752) and (C33269/A23215).

## Additional information

### Funding

| Funder | Grant reference number | Author |
|---|---|---|
| Cancer Research UK | C33269/A20752 | Maruf MU Ali |
| Cancer Research UK | C33269/A23215 | Maruf MU Ali |

The funders had no role in study design, data collection and interpretation, or the decision to submit the work for publication.

### Author contributions

Megan C Kopp, Conceptualization, Methodology, Validation, Formal analysis, Investigation, Resources, Data curation, Visualization; Piotr R Nowak, Methodology, Validation, Formal analysis, Investigation, Data Curation; Natacha Larburu, Validation, Investigation, Resources; Christopher J Adams, Validation, Resources; Maruf MU Ali, Conceptualization, Methodology, Validation, Formal analysis, Writing - original draft, Writing - review and editing, Visualization, Supervision, project administration, Funding aquisition

### Author ORCIDs

Maruf MU Ali (iD) https://orcid.org/0000-0001-5392-4029

### Decision letter and Author response

Decision letter https://doi.org/10.7554/eLife.30257.018
Author response https://doi.org/10.7554/eLife.30257.019

## Additional files

### Supplementary files

• Transparent reporting form
DOI: https://doi.org/10.7554/eLife.30257.011

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
