## [Decision Letter]

Thank you for submitting your article "in vitro FRET analysis of noncanonical Ire1 and BiP association and dissociation upon ER stress" for consideration by *eLife*. Your article has been reviewed by two peer reviewers, and the evaluation has been overseen by a Reviewing Editor and Randy Schekman as the Senior Editor. The reviewers have opted to remain anonymous.

The reviewers have discussed your study and formulated a consensus report.

The reviewers found your work potentially interesting, but indicated multiple technical points that need to be addressed in a revised version of the manuscript before we can proceed further.

Essential Revisions:

1) The key conclusion that Ire1-LD interacts with the ATPase of BiP and not with the SBD of BiP appears to rest mainly on the assumption that a C-terminally YFP-tagged BiP (BiP-YFP) would not result in a FRET signal with CFP-Ire1LD. A control experiment should be conducted with BiP-YFP to support this. Related to this problem: Why does the non-binding control give a FRET ratio of 0.34? Should this not be close to zero? It is considered fundamental that this point be rigorously addressed.

2) A critical concern relates to your claim that the Ire1LD is not recognized by BiP as a chaperone substrate, because the BiP substrate-binding domain mutant was not released from Ire1LD in the presence of unfolded proteins. If the same result could be obtained with a BiP ATPase deficient mutant, this claim would be strongly supported. Such additional experiments should be performed.

David Ron's group reported that BiP dissociated from the BiP-PERK complex in an ATP-dependent manner (NCB, 2000; PMID 10854322), suggesting that the BiP-PERK and BiP-IRE1a interaction is a chaperone-substrate interaction. Kenji Kohno's group also reported that yeast Ire1p associates with BiP as a chaperone substrate (MBC, 2003; PMID 12808051). These reports seem to not support your model and should be carefully discussed.

3) All pulldown experiments were done by pulling on BiP. Pull downs via Ire1 would be informative, because Ire1LD has a possibility to bind misfolded proteins.

4) Figure 4—figure supplement 1; the pulldown experiment showed that His-BiP still interacted with Ire1LD in the presence of TTRD18G, but there was completely no signal in the FRET experiment. This needs to be explained.

5) Figure 5 experiments; the data should be presented more carefully. The band intensity of Flag-Ire1 in lane 3 seems lower than in the other lanes. Please present the data of this experiment in a quantitative manner. Also, the experimental conditions should be described more precisely: which vector was used, and how much DNA was used in transfection?

6) It is unclear what exactly "total FRET pair emission" is. Please clarify. Is this the intensity in the 530 nm channel? If yes, please state this and specify the bandwidth at least of the detection.

7) It is unclear what exactly "FRET ratio" is. Is this the intensity in the 530 nm channel divided by 480 nm channel? Please clarify and give bandwidth.

8) Please define "inhibition constant", which should likely read "inhibition rate constant".

9) What is one sample? Is this the average of all 384 wells? Then six samples are fine. Otherwise, if six samples are six wells of a 384 well microplate this is unacceptable statistics.

[Editors' note: further revisions were requested prior to acceptance, as described below.]

Thank you for resubmitting your work entitled "in vitro FRET analysis of IRE1 and BiP association and dissociation upon Endoplasmic Reticulum stress" for further consideration at *eLife*. Your revised article has been favorably evaluated by Randy Schekman (Senior editor), a Reviewing editor, and two reviewers.

The manuscript has been improved but there are some minor remaining issues that need to be addressed before acceptance, as outlined below:

1) The "background" FRET ratio of about 0.34 is now well explained, but in the FRET field this is called "leakage" and can / should be subtracted before the FRET ratio calculation. Then the FRET ratio would be close to zero as expected. The authors at least should mention the word "leakage" somewhere, this would prevent misunderstanding.

2) The authors state in the discussion that ATP dependence is intimately associated with chaperone activity (Discussion section). I do not understand this argument.

3) Figure 5 caption could be a little more detailed – e.g. is the mutation indicated by the black six-ring?

---

## [Author Response]

Essential revisions:1) The key conclusion that Ire1-LD interacts with the ATPase of BiP and not with the SBD of BiP appears to rest mainly on the assumption that a C-terminally YFP-tagged BiP (BiP-YFP) would not result in a FRET signal with CFP-Ire1LD. A control experiment should be conducted with BiP-YFP to support this.

We would like to thank the reviewers for this excellent suggestion; upon reflection, this particular negative control experiment seemed to escape our attention and is superior to the one we employed – of just using YFP. The C-terminally placed YFP on BiP gave almost identical value as using our previous negative control (FRET ratio using YFP with CFP-IRE1 = 0.341; FRET ratio of Cterminal YFP on BiP = 0.346). This really strengthens our data since both negative controls give the same measurement; thereby, suggesting ~0.34 represents 0% FRET signal.

Related to this problem: Why does the non-binding control give a FRET ratio of 0.34? Should this not be close to zero? It is considered fundamental that this point be rigorously addressed.

The emission spectrum for CFP exhibits a maximum value at ~480nm when excited at 430nm wavelength; however, it has a secondary peak at 505nm with a broad shoulder that displays a significant intensity at 530nm wavelength (Author response image 1), the same wavelength that we measure YFP emission. The advantage of a broad shoulder is that it allows for a greater spectral overlap with the excitation peak of YFP, hence that is why CFP and YFP make an excellent choice for being FRET pairs (Author response image 2). The downside is that there is a significant CFP signal still present at 530nm wavelength; although there is an advantage to this too, since to visualise a FRET signal using this pair you need to have a significant FRET signal – otherwise you simply do not see FRET occurring above the background. For the benefit of the reviewers, we have conducted a spectral scan of CFPIRE1, and FRET interacting pair CFP-IRE1 and YFP-BiP, that measures fluorescence emission over the range 470 – 600nm upon excitation at 430-10nm (Author response image 3). Taking the value at 530/480 nm gives a FRET ratio ~0.33 – 0.34, which is the background reading for our assay. In summary, the 3-componemnt FRET UPR induction assay we develop in this paper gives excellent data, with clear signal and is highly reproducible. We believe this to be a valuable tool to further dissect the mechanism of UPR induction. We have added extra description to the Figure legend 1C to explain the ~0.34 background signal.

**Author response image 1. respfig1:** The predicted emission and excitation spectra for CFP when excited at 410 nm with bandwidth 10nm. The peak emission is measured at 480-10, but there is a significant signal present at 530nm, which is the peak emission point for YFP. The graph was generated by CLARIOstar plate reader BMG labtech.

**Author response image 2. respfig2:** The predicted emission and excitation spectra for CFP and YFP FRET pair when excited at 430-10 nm. YFP peak emission occurs at 530nm. The graph was generated by CLARIOstar plate reader BMG labtech.

**Author response image 3. respfig3:** A spectral scan of fluorescence emission intensity between 470nm and 600nm when excited at 430-10nm. The FRET signal peak at 530nm is clearly visible, also the significant background produced by CFP-IRE1 is observed, producing a background reading of ~0.34 at 530nm- 10/480-10nm wavelength.

2) A critical concern relates to your claim that the Ire1LD is not recognized by BiP as a chaperone substrate, because the BiP substrate-binding domain mutant was not released from Ire1LD in the presence of unfolded proteins. If the same result could be obtained with a BiP ATPase deficient mutant, this claim would be strongly supported. Such additional experiments should be performed.David Ron's group reported that BiP dissociated from the BiP-PERK complex in an ATP-dependent manner (NCB, 2000; PMID 10854322), suggesting that the BiP-PERK and BiP-IRE1a interaction is a chaperone-substrate interaction. Kenji Kohno's group also reported that yeast Ire1p associates with BiP as a chaperone substrate (MBC, 2003; PMID 12808051). These reports seem to not support your model and should be carefully discussed.

We have now conducted the experiment in the presence of 5mM ATP using the BiP ATPase deficient mutant, T229A, alone and in combination with the substrate binding mutant, V461F (see Figure 5—figure supplement 1); and have assessed the impact to the FRET signal upon addition of misfolded proteins. Using the ATPase deficient mutant produced exactly the same outcome as using just the V461F mutation alone. ATP did not affect dissociation between Ire1 and BiP. The primary determinant for dissociation was addition of misfolded protein (see Figure 5—figure supplement 1).

The reviewers have cited two articles that seem not to support our work. These studies form the basis of the competition model for UPR activation. In this model BiP binds to IRE1 via its substrate binding domain, which also is the site that misfolded protein binds to BiP. As the concentration of misfolded proteins increase, BiP is titred off IRE1 to activate UPR. But more importantly, the addition of ATP is sufficient to cause dissociation of BiP from IRE1 and not necessarily misfolded protein, making this process ATP dependent.

However, one could argue since binding occurs via the substrate binding domain of BiP, that the interaction is a chaperone substrate type interaction; this point is further emphasised by release being ATP dependent – ATP action is intimately associated with chaperone function.

Our in vitro data clearly indicates that binding between IRE1 and BiP occurs solely via the ATPase domain of BIP, and that this process is independent of ATP. Our model clearly suggests a UPR signalling productive interaction between BiP and IRE1 that’s dissociates upon a conformational change caused by misfolded protein binding to the substrate binding domain of BiP, and not the presence of ATP. Furthermore, since binding of BiP to Ire1, and detection of misfolded proteins, occurs upon different domains that are coupled together by allosteric change, there is no requirement for a competition to bind misfolded protein in order to activate UPR.

Regarding the role of nucleotides, it could be that nucleotides have influence over other aspects of UPR, it is known that nucleotides greatly affect the conformation of BiP, exactly how this relates to UPR signaling is an active area of ongoing research in our lab. We believe our model has great merit, and this is backed up by our careful in vitro analysis. We have now edited the discussion to include the points mentioned above (see Discussion section).

3) All pulldown experiments were done by pulling on BiP. Pull downs via Ire1 would be informative, because Ire1LD has a possibility to bind misfolded proteins.

The reason why we do not pull down on IRE1, is because BiP has a strong tendency to stick to the beads even in the absence of a tag, and thus the negative control sample contains a significant amount of BiP. We have observed this behaviour with other chaperone proteins including Hsp90. To overcome this, we simply tag and pull down on BiP. Un-tagged IRE1 LD does not stick to beads and hence gives a clean negative control, making the experiment interpretable. We have included an example experiment to show that un-tagged BiP sticks to beads as negative control. This is marked as* in Author response image 4.

**Author response image 4. respfig4:** Pull down assay, using his_6_-tagged IRE1 LD as bait and BiP as prey. The experiment does not work since un tagged BiP sticks to beads as negative control. The experiment works well when BiP is tagged and using IRE1 LD as prey, since untagged IRE1 LD does not bind beads.

4) Figure 4—figure supplement 1; the pulldown experiment showed that His-BiP still interacted with Ire1LD in the presence of TTRD18G, but there was completely no signal in the FRET experiment. This needs to be explained.

We would like to apologise to the reviewers; we have mistakenly presented the wrong gel. The gel was taken from an optimisation round with the wrong misfolded protein concentration. We have now presented the correct gel; furthermore, we have also increased the number of observations made for the FRET assay so that n=6.

5) Figure 5 experiments; the data should be presented more carefully. The band intensity of Flag-Ire1 in lane 3 seems lower than in the other lanes. Please present the data of this experiment in a quantitative manner. Also, the experimental conditions should be described more precisely: which vector was used, and how much DNA was used in transfection?

We have now quantified the signal in Figure 5. Also, we have added more description to the Materials and methods section.

6) It is unclear what exactly "total FRET pair emission" is. Please clarify. Is this the intensity in the 530 nm channel? If yes, please state this and specify the bandwidth at least of the detection.

We apologise for the lack of clarity -- we have now corrected this point by clearly stating the following:

“For our assay, we excited at 430nm wavelength with band width 10nm, and observed the fluorescence emission intensity at a 530nm wavelength with a bandwidth of 10nm, which was then divided by the emission intensity at 480nm also with band width 10nm, to give a FRET ratio (530-10/480-10nm), and this was used to measure signal output” (see Results section).

7) It is unclear what exactly "FRET ratio" is. Is this the intensity in the 530 nm channel divided by 480 nm channel? Please clarify and give bandwidth.

We have now clarified what the FRET ratio is with the same statement mentioned above.

8) Please define "inhibition constant", which should likely read "inhibition rate constant".

The inhibition constant, K_i_, is the binding constant relating to the interaction of a complex, with binding affinity K_d_, with the inhibiting molecule that is derived from the IC_50_ value (Martin et al., 2008). The K_i_ is more reflective of the binding affinity of the complex towards the inhibitor, whereas the IC_50_ is a measure of functional strength of inhibitor. The lower the value the better the affinity between complex and inhibitory protein. Also, IC_50_ can vary depending on substrate concentration used in the IC_50_ determination, whereas Ki is an equilibrium (and not a rate) constant and a better measurement for comparison.

We have defined the inhibition constant K_i_ in Figure legend 2.

9) What is one sample? Is this the average of all 384 wells? Then six samples are fine. Otherwise, if six samples are six wells of a 384 well microplate this is unacceptable statistics.

The majority of our experiments have now been conducted with n=6 or 6 independent experiments, with each independent experiment done in triplicate. Each independent experiment may have been from different protein preparations and are certainly done at different times. We have now ensured that all of our experiments are n=6 and in triplicate – except the IC_50_ determination which is n=3, because it requires so much protein – and we thank the reviewers for highlighting this point. So, for each point of data there is 18 observations made. It’s important to note that the protein samples are highly purified, concentrated, and homogeneous preparations done with meticulous care. Each protein expression and purification takes a lot of time and effort, with many rounds of expression and purification required to obtain a dataset. The signal from these experiments are superior to those conducted in cell based assays which may require many fold measurements to get statistically sound result due to the weak signal. In this assay, each measurement is highly significant, with a clear signal.

By way of a comparison, we have looked at similar experiments which have been done in vitro, using purified and concentrated proteins, in a 384 well plate format, using fluorescent intensity measurements and malachite green absorbance intensity (Preissler et al., 2015). Although, n=3 – 4 for these experiments, the number of repeats per independent experiment are not specified, but we can deduce based on the stock sample volume of 150μl for malachite green assay, the same volume as our stock sample, that they may be conducted in triplicate. Furthermore, the article that we have used as a reference to develop our in vitro FRET experiments, have used n=1, done in triplicate using 384 well plate format (Martin et al., 2008), producing beautiful in vitro FRET data. Thus, in vitro experiments using highly purified and concentrated protein produce clear FRET signal and data, whereas FRET signal in cells are much weaker and require more observations to compensate for this. Our assay produces excellent and reproducible data with a clear FRET signal being observed with each measurement. This is in line with other published studies conducting similar type of experiments.

[Editors' note: further revisions were requested prior to acceptance, as described below.]

1) The "background" FRET ratio of about 0.34 is now well explained, but in the FRET field this is called "leakage" and can / should be subtracted before the FRET ratio calculation. Then the FRET ratio would be close to zero as expected. The authors at least should mention the word "leakage" somewhere, this would prevent misunderstanding.

We have now edited the Figure legend 1C to incorporate the word ‘leakage’ with part of the figure legend stated below to highlight this point.

“The negative controls measure a FRET ratio of ~0.34 due to CFP contributing a significant fluorescence emission intensity at 530nm – also referred to as CFP leakage – when excited at 430nm. This allows for greater spectral overlap with YFP making CFP and YFP excellent FRET pairs, but adds to the background noise”.

2) The authors state in the discussion that ATP dependence is intimately associated with chaperone activity (Discussion section). I do not understand this argument.

We apologise for the rather cumbersome choice of words. We have now changed the sentence to read:

“However, since the interaction between IRE1 and BiP is mediated via the substrate binding domain, this suggests a chaperone substrate type interaction (Bertolotti et al., 2000; Kimata et al., 2003); this point is further emphasized by the process being ATP dependent – ATPase activity is an integral part of the Hsp 70 chaperone-substrate mechanism”.

The argument we make regarding ATP dependency is as follows:

Hsp70 chaperones are ATPase’s that hydrolyse ATP as part of its chaperone mechanism of action. The binding and release of BiP misfolded protein substrate is dependent on the interconversion of a low substrate affinity ATP bound state and a high substrate affinity ADP bound state. Thus, the ATPase activity of Hsp70 is an integral part of its chaperone mechanism.

A key observation for the UPR competition model is that the dissociation between IRE1 and BiP is dependent upon ATP, with ATP being sufficient to cause dissociation and not misfolded proteins. Since the dissociation of IRE1 and BiP is dependent upon ATP, it suggests that the interaction is same type of interaction as a BiP chaperone substrate interaction.

We show that the interaction between BiP and IRE1 is unaffected by the nucleotides, thus demonstrating that it has no relation to BiP chaperone-substrate mechanism, unlike the competition model.

However, the ATP dependence is a secondary point that suggests that our allosteric model is a UPR significant interaction. The main evidence is that interaction between IRE1 and BiP for the competition model occurs via the BiP substrate-binding domain, in exactly the same way that BiP chaperone-substrate interaction would occur. Whilst our model indicates IRE1 interaction to BiP occurs via BiP ATPase domain, suggesting a UPR significant interaction, not linked to chaperone-substrate function.

Aside from this point, the real strength in our model is that it offers a rationale to how UPR is sensitive to ER stress; particularly, as IRE1 has a low copy number in cells, whilst BiP is the most abundant protein in the ER. Competition between misfolded protein and IRE1 for binding to BiP substrate binding domain would require an enormous amount of misfolded protein to titre BiP off from IRE1 and therefore a competition model simply does not provide a mechanistically satisfactory explanation. In our model, binding of IRE1 to BiP, and binding of misfolded proteins, occur on two different domains that are allosterically linked, therefore there is no requirement for competition to activate UPR and provides an excellent rationale for a sensitive response.

3) Figure 5 caption could be a little more detailed – e.g. is the mutation indicated by the black six-ring?

We have added extra clarity to the figure by labelling the mutation and describing that no binding will occur in the presence of the mutation. Moreover, we have added extra detail to the Figure legend 5A that describes the figure better.